# Segmentation and Recognition of the Pathological Features of Squamous Cell Carcinoma of the Skin Based on Multispectral Imaging

**DOI:** 10.3390/jcm11133815

**Published:** 2022-07-01

**Authors:** Cheng Wang, Qi Chen, Tijie Gao, Shijun Guo, Huazhong Xiang, Gang Zheng, Dawei Zhang, Xiuli Wang

**Affiliations:** 1Institute of Biomedical Optics and Optometry, Key Laboratory of Medical Optical Technology and Instruments, Ministry of Education, University of Shanghai for Science and Technology, Shanghai 200093, China; shhwangcheng@163.com (C.W.); gtj_usst@163.com (T.G.); guosj@sumhs.edu.cn (S.G.); xiang384524@163.com (H.X.); gangzheng@usst.edu.cn (G.Z.); 2Institute of Photomedicine, Shanghai Dermatology Hospital, School of Medicine, Tongji University, Shanghai 200443, China; chenqi235533@163.com; 3Engineering Research Center of Optical Instruments and Systems, Ministry of Education, Key Laboratory of Modern Optical Systems, Shanghai University of Technology, Shanghai 200093, China

**Keywords:** cutaneous squamous cell carcinoma (cSCC), histopathology, multispectral imaging, image segmentation, feature recognition

## Abstract

Cutaneous squamous cell carcinoma (cSCC) is one of the most common skin cancers, a definitive diagnosis of cSCC is crucial to prevent patients from missing out on treatment. The gold standard for the diagnosis of cSCC is still pathological biopsy. Currently, its diagnostic efficiency and accuracy largely depend on the experience of pathologists. Here, we present a simple, fast, and robust technique, a microscopic multispectral imaging system based on LED illumination, to diagnose cSCC qualitatively and quantitatively. The adaptive threshold segmentation method was used to segment the multispectral images into characteristic structures. There was a statistically significant difference between the average nucleocytoplasmic ratio of normal skin (4.239%) and cSCC tissues (15.607%) (*p* < 0.01), and the keratin pearls cSCC have well-defined qualitative features. These results show that the qualitative and quantitative features obtained from multispectral imaging can be used to comprehensively determine whether or not the tissue is cancerous. This work has significant implications for the development of a low-cost and easy-to-use device, which can not only reduce the complexity of pathological diagnosis but can also achieve the goal of convenient digital staining and access to critical histological information.

## 1. Introduction

Cutaneous squamous cell carcinoma (cSCC) is one of the most common skin cancers, accounting for 20% to 50% of cutaneous malignant tumors [1,2]. It develops more frequently in people exposed to environmental factors such as ultraviolet light, as well as smoking, chronic infections, and immunosuppression [3]. The incidence rate of cSCC increases worldwide along with the population aging, posing a serious human health risk. Given the high mortality and morbidities associated with this type of cancer, a correct early diagnosis is very helpful in the management of this condition [4,5].

Traditionally, pathological diagnosis is the gold standard, according to the tissue structure and cytological features. However, the diagnosis process is complicated and largely relies on the pathologist’s experience, and is also subject to intra- and inter-observer variability [6]. Additionally, the traditional microscopy imaging technique usually presents an RGB image that can be conceived as a multispectral image with only three spectral bands, so some information in the image is mixed. The pathologist spends a great deal of time making an accurate decision by observing histological samples. Furthermore, as samples stained by different laboratories may have different degrees of uneven staining or over-straining, this may lead to observer variability and also makes it difficult to extract parameters. Therefore, a new method is necessary to mitigate the disadvantages.

The interaction between light and biological tissue involves scattering and absorption of the photon. Multispectral imaging (MSI) is an optical spectroscopy imaging modality that directly measures the incoming radiance spectra of light [7]. There are two major detection modes, depending on the incidence of light within the tissue: light reflection or light transmission. The spectral information measured by these technologies is usually related to the information about both scattering and absorption of light within the sample. MSI technology has a unique capability for skin characterization because it can take advantage of the spatial relationships among the different tissue absorption spectra in a neighborhood. Spectral data cube analysis can incorporate complex spectral–spatial models that provide a more accurate classification of image features specific to a targeted disease [8]. The histopathological images show morphologic changes associated with cutaneous malignancy, such as irregular cell proliferation, irregular nuclear deformation, etc. In the meantime, the composition of the tissue also changes, with collagen and lipid levels varying widely between normal tissue and skin tumors. The optical properties of tissues are also altered by these changes in morphology and composition, which makes spectral imaging of great significance in tissue recognition. Spectral imaging enables simultaneous spectral and image analysis, providing information about morphology and composition on a single graph. Therefore, the samples can be analyzed by using the spectral characteristics and spatial information contained in the multi-band image.

In recent years, MSI technology for pathological diagnosis has been used in studies on digital staining in increasing numbers. S Ortega et al. have reviewed systematically that the use of MSI suggested an improvement in the detection of diseases and clinical practice and brought new opportunities in the analysis of histological samples [9]. Nevertheless, the number of studies in this field is currently limited, and more research is needed to confirm the advantages of this technology compared to conventional imagery. Additionally, to solve the problem of different sample staining in different laboratories, Yagi et al. proposed a color normalization method based on MSI technology and standard color tiles, improving the accuracy and consistency of diagnosis in different laboratories [10]. Bautista et al. stained sections digitally based on the spectral information of different tissues in MSI, which highlight fibrous structures and emphasize the less remarkable structures in hematoxylin and eosin (H&E) stained sections without additional dyeing operations [11]. What is more, with MSI, Bayramoglu et al. used conditional generative adversarial networks to digitally stain unstained lung tissues and extract spectral information to mimic the appearance of H&E staining, directly eliminating the staining steps [12].

Though MSI technology has been widely used in many fields, there are few studies about its application in cutaneous pathology. To explore the potential clinical value of MSI technology, this study, based on a self-developed microscopic MSI system, used the optical absorption properties of H&E staining to analyze pathological sections of cSCC and normal skin tissue, tested digital segmentation’s actual imaging effect and ability, and is expected to provide a new method for the pathological analysis of skin and skin cancers.

## 2. Materials and Methods

### 2.1. Experimental Equipment

We used the narrow-band LED illumination-based microscopic MSI system from previous studies [13]. Transmission and reflection are the two main imaging modes available, depending on the characteristics of biological tissues. Transmission microscopes work by allowing light to penetrate the image of a sample illuminated by light and are suitable for transparent or thin samples where the light source is usually not on the same side as the detector. Reflective microscopy can be used on both transparent and opaque samples by illuminating them with light falling from above and illuminating them with a light source through an objective to obtain a microscopic image of the sample surface. The light source of the fallout illumination is usually on the same side as the detector, which allows for smaller systems, simpler optical paths, and easier design of multiple narrowband light sources. The subject of this study is skin tissue. Reflection imaging is more similar to in vivo skin imaging modality, which makes future system modification easier. In summary, if the multi-channel LED illumination is used, the reflective microscope is easier to design the light path, has a wider application range, and is more suitable for the multispectral microscopy system proposed in this experiment. The working principles are shown in Figure 1. The system mainly uses 13 narrow-band LED lights which the control software can light in sequence; the fiber bundle and the coupling lens group shape the light emitted by the LEDs into parallel beams. This is then reflected by the beam splitter onto the objective lens and focused on the sample. The light reflected back from the sample is focused through the imaging lens onto a monochromatic digital camera’s sensor, and the camera control program is used to sequentially expose and acquire spectral images which form a sequence that leads to simultaneously acquiring both the image and the image’s spectral information, and any one of the pixels in the image sequence can be taken for image processing and spectral analysis. In order to study the basic performance of our system’s relevant functions, we used a fiber optic spectrometer (USB2000+, Ocean Optics, FL, USA) and a standard discrimination rate board (USAF1951, Thorlabs China, Shanghai, China) to evaluate the system, and the performance parameters are shown in Table 1.

### 2.2. Experimental Samples

A total of 10 female, hairless, immunocompetent SKH-1 mice (12 weeks old) were provided by Shanghai Public Health Clinical Center, Fudan University (Shanghai, China). The skin condition of cSCC was induced by solar-simulated UVR (Solar UV Simulator, SIGMA, Shanghai, China) five times weekly. The initial minimal erythema dose (MED) was 160 mJ/cm^2^ for UVB and 2520 mJ/cm^2^ for UVA. Hairless mice were irradiated with 90% of the MED in the first and second weeks. The thickness of the mouse skin gradually increased over time, and the tolerance to UVR increased accordingly. Therefore, the dose was increased to 100% of the MED in the third week and then increased by 12.5% MED every week. By the eighth week, the UVB dose was increased to 260 mJ/cm^2^ per day and was maintained thereafter. Irradiation was stopped when papules measured equal to or more than 2 mm in diameter [14]. Subsequently, the papules gradually grew to various sizes of cauliflower-type lesions. When the tumor grew to 5 mm in diameter, 10 normal mice and cSCC mice were sampled. The tumors of anesthetized cSCC mice were completely excised with a surgical blade, and the skin of the same position on the back of normal mice was taken for sampling. Then the tumor tissue and normal tissue were made into pathological sections to observe the histological changes in skin lesions.

All animal procedures complied with protocols approved by the Institutional Animal Care and Use Committee of Tongji University.

### 2.3. Data Acquisition

For this system, the acquisition of data was done under the guidance of a pathologist in a dark room without any interference from ambient light. When acquiring images, a 13-band grayscale image was collected for each lesion. A histological slide was placed on the multispectral microscope’s translation stage, and the light source was switched to both capture and store images which ensured that the camera exposure time was the same throughout the process of illuminating different areas of the sample with the same wavelength of light. Noise correction was performed on the collected spectral images using the dark current and radiation between spectral bands. First, the camera’s dark noise was collected without illumination, and then the light source was switched so that the image in the unsampled area of the slide provided the values for reference light intensity. Images of the sample being tested were collected at each wavelength under the same lighting conditions as the sample light intensity. The reflectance correction principle is shown in Formula (1).
(1)$$R\left(\lambda \right)=\frac{I_{raw}\left(\lambda \right)-I_{dark}}{I_{ref}\left(\lambda \right)-I_{dark}}$$

In the formula, $I_{raw}\left(\lambda \right)$ represents the sample image’s light intensity, $I_{dark}$ represents background light intensity (when there is no lighting), and $I_{ref}\left(\lambda \right)$ represents the light intensity of the reference image on optical slides of different wavelengths. After taking all the sample images, a program batch processes the images of all 13 bands and saves the combined image gray value as the reflectance. The average reflectance of the sample’s region of interest can be extracted, as can the reflectance of a certain point. At the same time, although problems such as unavoidable noise and uneven illumination may appear in the image, the average processing method is used to remove image noise. Specifically, this is done by switching the light source to collect five images in the region of interest continuously. Then, the image preprocessing algorithm averages the pixel values of each group of five images, thus reducing interference caused by random noise. To correct uneven illumination of the image, we have adopted an adaptive two-dimensional gamma function method which can better correct image uniformity [15], brighten overly dark places, darken overly bright places, and improve the image’s overall contrast and clarity.

### 2.4. Segmentation Method

A total of 5 images in each histological sample of 13 wavelengths for 65 multispectral grayscale images were taken. Multispectral grayscale images can use spectral dimensions to accurately display the morphological characteristics of different tissues. Because of the huge variation in gray value, highlighted features are usually obvious, and there is no need to use complex algorithms for identification and extraction. However, the image quality of MSI is related to the dyeing level of the tissue section and light source, so the image may have a lot of noise and uneven local brightness. Firstly, local histogram equalization was carried out to improve the contrast of the image and make the nucleus more obvious. Secondly, a median filtering operation was carried out to remove the salt and pepper noise on the image. Finally, due to the large number of nuclei and the existence of fuzzy boundary areas, if we want to segment them accurately, a lot of time should be spent marking them manually. Therefore, using adaptive threshold segmentation, the image was divided into many regions, and the threshold segmentation was carried out in each region. Although it cannot be accurately segmented, due to the same operation of each image, the obtained nucleus had the same standard, the calculated nucleocytoplasmic ratio can be compared to each other, and the segmentation speed is fast [16]. In this study, the selected spectral bands are 520 nm, 600 nm, and 660 nm.

### 2.5. Statistical Analysis

All measurement values of nucleocytoplasmic ratio were expressed as average, maximum and minimum. Comparisons between two groups were performed using Student’s *t*-tests. GraphPad Prism 7 software was used for analyses, and statistical significance was defined as *p* < 0.05.

## 3. Experimental Results and Analysis

### 3.1. UVR Induced Skin Canceration in Pathology

Following ultraviolet ray (UVR) exposure, various sizes of cauliflower-type lesions were found on the previously smooth skin of SKH-1 mice, then the tumor tissue and normal tissue were made into pathological sections to observe the histological changes of the skin tumor. This tumor was well-differentiated; the disorderly growth of the squamous epithelial cells in these large nests with pink keratin was seen in pathology after UVR exposure. Pathological examination revealed characteristic pathological changes of cSCC. The normal skin and cSCC tissue pathological images are shown in Figure 2A,B.

### 3.2. The Features on the cSCC Images Were Highlighted after Segmentation

A total of 5 images in each histological sample of 13 wavelengths for 65 multispectral grayscale images were taken by microscopic MSI system based on LED illumination. Multispectral grayscale images can use spectral dimensions to accurately display the morphological characteristics of different tissues. Figure 3 shows the single-band and pseudo-color composite images of the mouse skin’s normal and cSCC tissue areas, with 3A and 3B displaying the single-band and composite pseudo-color images of the cSCC tissue slide and the normal tissue slide, respectively. It can be seen from the segmentation results that the structural features of normal tissue and cancerous tissue in each band of images are better segmented. Using a certain difference in the depth of staining to show both the dermis with and without lesions, the segmentation results can highlight what would have been initially inconspicuous features on the pathological image while avoiding influence from staining. The image at 660 nm spectral bands is used to segment cell nuclei; the image at 600 nm spectral bands is used to segment large lipid droplets in normal tissues and keratin pearls in cancerous tissues; and the image at 520 nm spectral bands segments collagen fibers in normal tissues and diseased epidermis in cancerous tissues. Corresponding to the characteristics of typical cSCC and according to the pseudo-color image synthesized from the segmentation results, the nuclei of the cancerous tissue have become larger and darker. The keratin pearls (a marker of cSCC) have appeared. Seeing the various structures made more evident by the pseudo-color image, the cancerous histopathology can be well distinguished from a qualitative point of view, and the goal of digital staining is achieved. The adaptive threshold method has a significant reference value because it can better extract various structures from multispectral images, avoid the different diagnostic decisions caused by staining differences, and provide pathologists with a more objective and more realistic basis for diagnosis.

### 3.3. Nucleocytoplasmic Ratio Quantitative Analysis Can Distinguish Obscure cSCC from Normal Skin after Segmentation

A quantitative analysis of the nucleocytoplasmic ratio can be done for qualitatively analyzed histological samples with hidden keratin pearls. The resolution of the raw image is 1280 × 1024 pixels. The area with 100 × 100 pixels is about one-tenth of the whole image, which is good for selecting multiple areas and reducing the probability of area overlap. Moreover, if the selected area is too large, it is likely to contain blank non-nuclear areas of the image, and if it is too small, it is likely to have specific areas (blank or aggregated areas of nuclei), which will affect the final recognition. In this study, we used sliding window technology to create a 100 × 100 pixels window to search the region of interest and further analyze the nuclei of the normal and cancerous tissues. Limiting the threshold value of the searched area, underexposed images and the junction area between the nucleus and keratin pearls were excluded, and the nucleus-cytoplasmic ratio was calculated for the automatically selected area. Finally, five windows were selected to calculate the nuclear-cytoplasmic ratio in both normal and cancerous tissues. As shown in Figure 4, Figure 4A is a normal tissue cell with an 8.99% average nucleocytoplasmic ratio in five regions. Figure 4B is cancerous tissue with a 25.9% average nucleocytoplasmic ratio in five regions. It can be seen from this that normal tissue’s nucleocytoplasmic ratio is less than that of cancerous tissue and that quantification of the nucleocytoplasmic ratio of tissue sections allows for the automatic differentiation of normal and cancerous tissue. To statistically identify the differences between normal and cancerous tissue, 21 sections of normal and cancerous tissue were segmented and calculated their nucleocytoplasmic ratio. There is a significant difference (*p* < 0.001) in the nucleocytoplasmic ratio of normal and cancerous tissues, with the results in Figure 5 and the statistical results in Figure 6. During the early stages of carcinogenesis, there may be cases where keratinization of the carcinoma is not yet obvious. Therefore, to further analyze the difference between normal and cancerous tissues by simply calculating the nucleocytoplasmic ratio, we analyzed the specificity and accuracy with average, maximum, and minimum nucleocytoplasmic ratio values with the ROC curve results, shown in Figure 7.

## 4. Discussion

Although multispectral imaging has been widely used in fluorescent labeling research and has become a commonly used information enhancement method in research since the invention of multicolor fluorescent labels, the design of fluorescence-based multispectral imaging systems requires full consideration of spectral disturbances. In practice, the excitation light’s spectral width is highly required, which should be narrow enough [17]. This paper studied the segmentation of different structures of conventional, single H&E stained cSCC tissue via a self-developed multispectral microscopy imaging system with narrow-band LED illumination. The optical properties of the tissue reflected by images of different wavelengths could effectively identify the tissue’s various structures. The image in an H&E stain slide is similar to the stained image by other additional operations. The MSI helps achieve digital staining abilities, improve the visual effects, and simplify the steps, providing a basis for studying the cSCC pathology. According to this study, with the application of MSI in the diagnosis of cSCC, the equipment cost and slice process are reduced, and the manpower required for pathological diagnosis is also reduced. Additionally, the problem of diagnostic differences caused by differences in tissue staining is solved. All the above indicated that the MSI process is advantageous for the diagnosis of cSCC.

However, this system is not automated enough. During our experiments, because the system is still semi-manual, we encountered many common problems that optical microscopy imaging has, such as issues caused by blurry images, under or overexposure, uneven lighting, or a lack of focus. Although these issues are operationally important, they can be solved by some technical means (such as the automatic focusing technology realized by combining the electronic control platform and the image focusing algorithm [18]). However, they mean that one of the main goals of this field of research in the future is cooperation with automatic control algorithms and driving devices to achieve automatic focusing, field scanning, and image stitching. If that is done alongside the analysis function, it will be possible to create a set of special equipment for pathological analysis truly. Additionally, looking at things from the point of view of the light source being utilized, narrow-spectrum excitation and analysis capabilities are required for many multicolor fluorescently labeled pathologies. Therefore, from the emission spectrum of the LED light source selected in this system, we can see that as long as the four light sources in the range of 520–570 nm are limited by the spectral width and the lower half-width of the excitation spectrum of all 13 channels is within 20 nm, it may also be suitable for some pathological sections of multicolor fluorescent labeling with a fluorescence emission peak lower half-width of 30–60 nm.

The establishment of a noninvasive diagnosis system for skin tumors is an important direction of modern dermatology. Optical coherence tomography (OCT) and Reflectance confocal microscopy (RCM) are emerging noninvasive techniques. OCT is a popular technique that, although not yet a routine technique, is being used in several specialized dermatological practices and hospitals. OCT penetrates the skin up to 1.5 mm, and it has been mainly used for diagnostics of basal-cell carcinoma [19,20] and presurgical margin assessment of non-melanoma skin cancer [21]. However, OCT has a low resolution, and cell-level imaging is difficult. RCM has a high resolution of 1 um, but its low penetration depth makes it difficult to observe information in the deep dermis.

Pathological biopsy is still the gold standard of diagnosis, focusing on solving the diagnostic efficiency and the diagnostic differences caused by staining; this paper verifies the ability of our self-designed multispectral microscopy imaging system in the digital staining of cSCC tissue sections and the segmentation of the main pathological features. By analyzing the inherent spectral characteristics of more skin diseases in the visible light range (from both image and spectral aspects), studying more excellent feature extraction algorithms for the extraction of typical pathological features of spectral or feature channel images, and realizing digital staining on unstained sections, our future research will be in the direction of further system automation and artificial intelligence. After establishing more animal models or collecting more clinical data, accumulating multispectral data of other diseases, and solving the pathological diagnosis of more diseases, we expect to make more efficient, high-quality diagnoses with an optimal overall cost.

## 5. Conclusions

In the study, our self-developed LED illumination-based microscopic multispectral imaging system was utilized, and we obtained pathological multispectral images of both normal skin tissue and squamous cell carcinomas. Usually, there are cancer nests in the tissues of squamous cell carcinoma, but because of the size limitation of the image, it is very likely that no cancer nests can be found on one image, so it is necessary to calculate the nuclear-cytoplasmic ratio of the cells. Due to the uneven illumination and different degree of slice dyeing, the edge area of each image is dark, and there is a certain gray difference between different images; it is necessary to denoise and histogram equalize the image. The adaptive threshold segmentation was used to segment the multispectral images into structures such as cell nuclei, collagen, and keratin pearls and allowed us to effectively extract important diagnostic information. Making use of sliding window technology, we created windows 100 × 100 pixels and searched randomly in the image, preserving five eligible areas. The nucleocytoplasmic ratio was calculated within the window, after which a ROC curve was used to analyze sensitivity and specificity. At a *p* < 0.01, our statistically significant results showed that the average nucleocytoplasmic ratio of normal skin tissue was 4.239%, and the average nucleocytoplasmic ratio of squamous cell carcinoma tissue was 15.607%. When the maximum value and the average value are used as the criterion, the AUC of the ROC curve is 1, so it has good sensitivity and specificity. Additionally, because squamous cell carcinoma’s keratin pearls have well-defined qualitative features, the qualitative and quantitative features of keratin pearls can be used to comprehensively determine whether or not the tissue is cancerous.

## Figures and Tables

**Figure 1 jcm-11-03815-f001:**
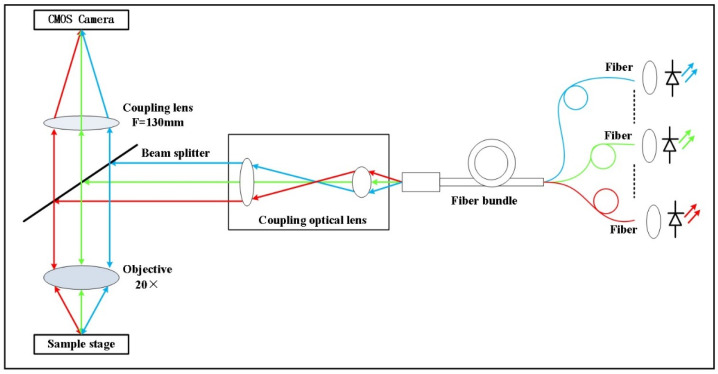
Optical schematic diagram of the system.

**Figure 2 jcm-11-03815-f002:**
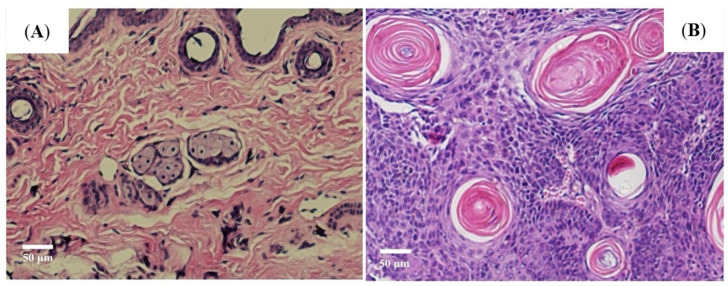
Histological staining of mice exposed to UVR. (**A**) Normal skin; (**B**) cSCC.

**Figure 3 jcm-11-03815-f003:**
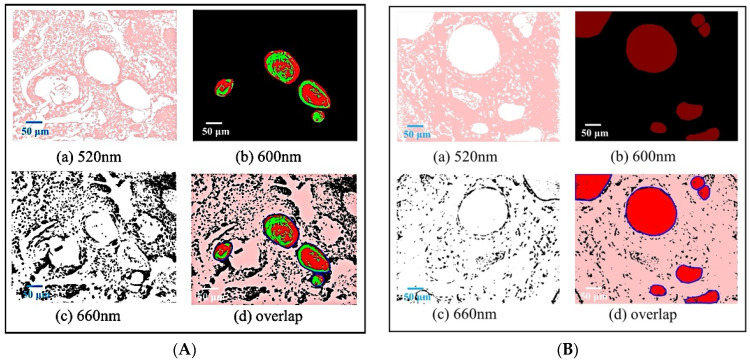
Single-band images of stained sections of normal and cancerous tissues (scale bar 50 µm). (**A**) cSCC tissue; (**B**) normal skin tissue; (**a**) Image at 520 nm spectral bands; (**b**) Image at 600 nm spectral bands; (**c**) Image at 660 nm spectral bands; (**d**) mixed imaging.

**Figure 4 jcm-11-03815-f004:**
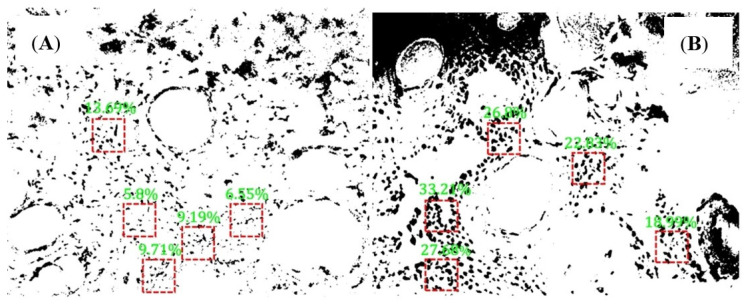
Comparison of the nucleoplasmic ratio between normal tissue sections and cSCC tissue sections. (**A**) the nucleoplasmic ratio of normal tissue; (**B**) the nucleoplasmic ratio of cSCC cancerous tissue.

**Figure 5 jcm-11-03815-f005:**
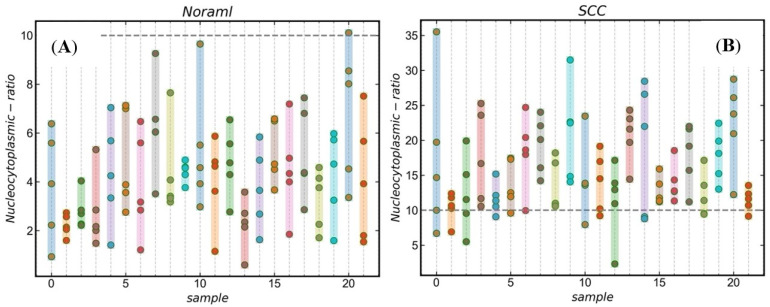
Distribution of the nucleoplasmic ratio of skin tissue cells. (**A**) Normal; (**B**) cSCC.

**Figure 6 jcm-11-03815-f006:**
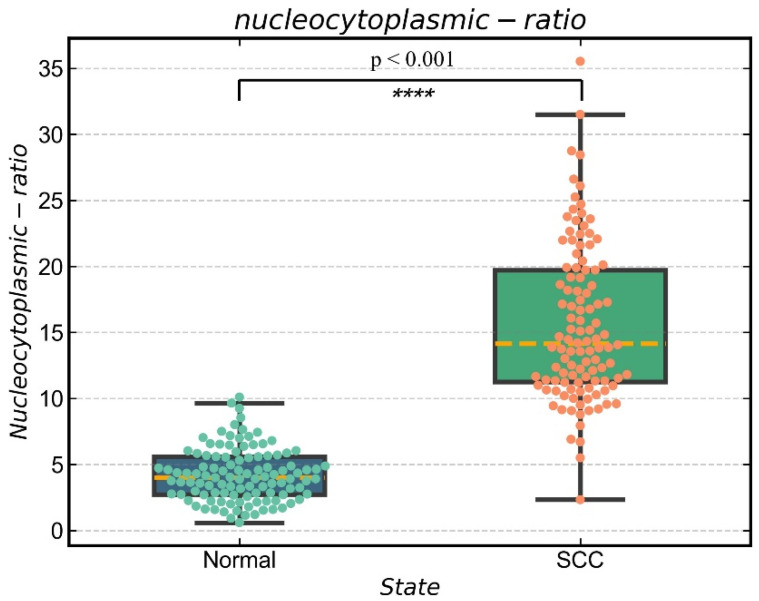
Values of nucleocytoplasmic ratio were expressed as average, maximum and minimum. Comparisons between two groups were performed using Student’s *t*-tests; **** *p* < 0.05.

**Figure 7 jcm-11-03815-f007:**
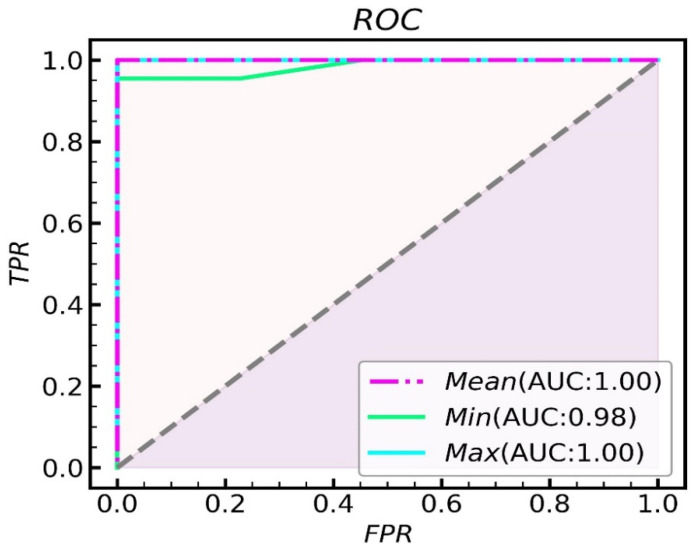
ROC curve of recognition rate.

**Table 1 jcm-11-03815-t001:** System performance parameters.

Number of Wavelengths	Wavelength (nm)	Spectral Resolution (nm)	Spatial Resolution (μm)	Field of View (μm)	Magnification
13	420–670	20	≤0.4	520 × 416	140

## Data Availability

Data are available on request to the corresponding authors.

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
