# Peer review of "Segmentation and Recognition of the Pathological Features of Squamous Cell Carcinoma of the Skin Based on Multispectral Imaging"

_jcm, 2022, doi:10.3390/jcm11133815_

Round 1
Reviewer 1 Report
An interesting original article about the use of a microscopic multispectral imaging system based on LED illumination in the diagnosis of cutaneous squamous cell carcinoma, showing a statistical significant difference between the average nucleocytoplasmic ratio of normal skin (4.239%) and cSCC tissues (15.607%), showing that this device may be useful to pathologist in the diagnosis of this condition; only minor queries:
In the introduction, you should add: "given the high mortality and morbidities associated with this type of cancer, a correct early diagnosis is very helpful in the management of this condition" and cite: doi: 10.3390/curroncol28040213. and doi: 10.3390/medicina57060563.
Thank You
Reviewer 2 Report
Abstract: At the end of abstract, the authors claim: “[…]but can also achieve the goal of convenient digital staining and access to critical hematological information. What do you mean? I think that the authors wanted to mean “histological information”. Please, correct.
Introduction: “cytopathic characteristics” is wrong. In histopathology it is better to say “cytological features”. Please, correct.
“However, the diagnosis process is complicated and largely relies on the experience of the pathologist, resulting in diagnostic inefficiency”. This sentence is not only wrong, it is also deeply offensive. Authors should remember that histopathology is the basis for diagnosing any skin lesion. Furthermore, their research work is based on training an optical segmentation method starting from histo / cytopathological characteristics. Therefore, please change this wrong sentence with a calmer, more factual statement. The pathologist's experience does not invalidate the inefficiency of the diagnosis! And, furthermore, I remind the authors (I deal personally with Artificial Intelligence applied to digital pathology) that any trained algorithm is always subjected to validation by a pathologist.
“The doctor always spends a great deal of time making an accurate decision by observing some histological samples”. Not a generic doctor, but pathologist. Please, correct!
“Furthermore, as samples stained by different laboratories may have different degrees of uneven staining or over-staining, it is impossible to obtain consistent diagnostic results”. I disagree with the authors. I understand very well that in order to strengthen the usefulness of the study, the authors need to highlight the limits of traditional diagnostic methods, but it is not true that routine histopathology is unable to ensure consistent diagnostic results. I would have understood much more if the authors had talked about melanocytic pigmented lesions of the skin, where, in fact, there is a big interobserver disagreement, but, although there is also in the case of other cancers such as cSCC, this is not at all. high enough to make the authors say that consistent diagnostic results are not obtained. Please, be more honest and rephrase the whole concept.
The rest of the introduction is very clear. All right. I would suggest to the authors to add some very small information on the epidemiology of cSCC perhaps referring to this recent and beautiful paper:
Cives M, Mannavola F, Lospalluti L, Sergi MC, Cazzato G, Filoni E, Cavallo F, Giudice G, Stucci LS, Porta C, Tucci M. Non-Melanoma Skin Cancers: Biological and Clinical Features. Int J Mol Sci. 2020 Jul 29;21(15):5394. doi: 10.3390/ijms21155394. PMID: 32751327; PMCID: PMC7432795.
Material and methods: This section is clear and I have no comments.
Results: There are some problems. The picture is not good. Indeed, in fig.2A the authors claim to present “normal skin” but in that image there is only derm with a few apocrine glands. Please, replace it with an appropriate picture. Furthermore, the authors have to improve the quality (c.d. resolution) of the images.
Same thing for Figure 3A-B and for Figure 4. Please, improve the quality.
Discussion and conclusions are good enough. Please, the authors should to reverse the two sections: before Discussion and then Conclusions.
Reviewer 3 Report
Thank you very much for sending us your manuscript. The listed work deals with the exciting topic of detecting pathological formations of skin tumors by means of multispectral imaging and identifying tumors in this way.
Please state conflict of interest in the manuscript, as well as a financial disclosure
has an animal test application been submitted?
Some questions still remain:
-please describe the resolution of the recordings as well as the possible depth of the recording also in the in-vivo model
-Please explain to what extent histopathological biopsies could then be omitted?
-Even if no tumor cells could be identified superficially, would no further surgical biopsies be performed? You will have to go into this in detail, since this is exactly what is at stake in perspective.
-Are there limiting factors for the in vivo assessment?
-In the case of skin tumors, it is often not even clear at first which tumor it is. can the etnity of the tumor also be differentiated with your methodology?
-How deep can an examination be performed with the device?
-I would recommend that alternative imaging options such as OCT are also discussed in the introduction or in the discussion.
-In the work mentioned, ex vivo preparations are examined that have already been cut, which would not be possible in vivo? please elaborate on this
Please consider to also include and discuss following publication regarding imaging of tumor:
Optical coherence tomography for presurgical margin assessment of non‐melanoma skin cancer–a practical approach
SA Alawi, M Kuck, C Wahrlich, S Batz, G McKenzie, JW Fluhr, ... Experimental dermatology 22 (8), 547-551 Assessment of a scoring system for Basal Cell Carcinoma with multi‐beam optical coherence tomography C Wahrlich, SA Alawi, S Batz, JW Fluhr, J Lademann, M Ulrich Journal of the European Academy of Dermatology and Venereology 29 (8), 1562-1569 Correlation of optical coherence tomography and histology in microcystic adnexal carcinoma: a case report SA Alawi, S Batz, J Röwert‐Huber, JW Fluhr, J Lademann, M Ulrich Skin Research and Technology 21 (1), 15-17 Page 1 Introduction "diagnosis process is complicated and largely relies on the experience of the pathologist, resulting in diagnostic inefficiency" in my experience, the pathological evaluation of specimens is very effective, and these are also reviewed by other experienced collaborators. if tumor specimens are involved, experienced pathologists usually receive the specimens anyway. please respond to this, but i would not let the sentence stand as it is. Method/Material How high is the sensitivity and specificity of the examination? this is not addressed at all. Discussion the clinical consequence as well as the weaknesses of the examination method must also be addressed here. in my opinion, simply saying that the pathology is inefficient and that new approaches are necessary here is not enough. How high is the sensitivity and specificity of the examination? this is not addressed at all.Author Response
Please see the attachment.

Round 2
Reviewer 2 Report
The authors have corrected all the points highlighted. Manuscript can be accepted.